# The Use of Flavylium Salts as Dynamic Inhibitor Moieties for Human C*b*_5_R

**DOI:** 10.3390/molecules28010123

**Published:** 2022-12-23

**Authors:** Oscar H. Martínez-Costa, Laura Rodrigues-Miranda, Sofia M. Clemente, António Jorge Parola, Nuno Basilio, Alejandro K. Samhan-Arias

**Affiliations:** 1Departamento de Bioquímica, Universidad Autónoma de Madrid (UAM), C/Arturo Duperier 4, 28029-Madrid, Spain; 2Instituto de Investigaciones Biomédicas ‘Alberto Sols’ (CSIC-UAM), C/Arturo Duperier 4, 28029-Madrid, Spain; 3Laboratório Associado Para a Química Verde (LAQV), Rede de Química e Tecnologia (REQUIMTE), Departamento de Química, Faculdade de Ciências e Tecnologia, Universidade NOVA de Lisboa, 2829-516 Caparica, Portugal

**Keywords:** flavonoids, chalcones, hemiketal, flavylium, quinoidal base, cytochrome *b*_5_ reductase

## Abstract

Cytochrome *b*_5_ reductase (C*b*_5_R) is a flavoprotein that participates in the reduction of multiple biological redox partners. Co-localization of this protein with nitric oxide sources has been observed in neurons. In addition, the generation of superoxide anion radical by C*b*_5_R has been observed. A search for specific inhibitors of C*b*_5_R to understand the role of this protein in these new functions has been initiated. Previous studies have shown the ability of different flavonoids to inhibit C*b*_5_R. Anthocyanins are a subgroup of flavonoids responsible for most red and blue colors found in flowers and fruits. Although usually represented by the flavylium cation form, these species are only stable at rather acidic pH values (pH ≤ 1). At higher pH values, the flavylium cation is involved in a dynamic reaction network comprising different neutral species with the potential ability to inhibit the activities of C*b*_5_R. This study aims to provide insights into the molecular mechanism of interaction between flavonoids and C*b*_5_R using flavylium salts as dynamic inhibitors. The outcome of this study might lead to the design of improved specific enzyme inhibitors in the future.

## 1. Introduction

NADH-Cytochrome *b*_5_ reductase (C*b*_5_R) is a flavoprotein that reduces multiple biological redox partners [1,2,3,4,5,6,7]. One of these acceptors is ferricytochrome *b*_5_ (C*b*_5_), which C*b*_5_R reduces by using NADH in a one electron-dependent reduction reaction [8,9]. Two forms of C*b*_5_ have been described: a soluble and a membrane-bound form [8]. C*b*_5_ reduction by C*b*_5_R is prompted by the existence of C*b*_5_R isoforms co-localizing with C*b*_5_ isoforms [6,10].

Co-localization of this protein with nitric oxide sources has been observed in neurons [11,12]. In addition, the existence of some relatively novel redox partners in nitric oxide signaling and vascular function has been shown [13,14]. In addition, the formation of oxygen free radicals, such as superoxide anions, by reductase in the absence of electron acceptors has been described [1,10]. This radical production is stimulated in the presence of alternative electron acceptors of C*b*_5,_ such as cytochrome *c* [15], or compounds, such as heterocyclic amines, which are potent carcinogenic agents [16]. Tight regulation of this enzyme must exist to avoid an eventual generation of both radicals (nitric oxide and superoxide anion), which production might lead to deleterious effects in cells and tissues [17]. A search for specific C*b*_5_R inhibitors to understand the role of this protein in this function has been initiated [13].

Flavonoids are natural polyphenols present in flowers, fruits, and other parts of some plants [18]. They are low molecular weight compounds having several beneficial health effects, such as antioxidant, anti-inflammatory, anti-mutagenic, and anti-carcinogenic properties [18]. These substances are derived from the secondary metabolites of plants and participate in various biological functions of plants [19,20].

Flavonoids are divided into subgroups (Figure 1), depending on the position of the B-ring and the degree of unsaturation and oxidation on the C-ring [19]. Some studies have shown that the inhibitory ability of some flavonoids, such as flavones, flavonols, flavanones, flavan-3-ols, and isoflavones, on enzymes depends on structural factors, such as double bonds between C-2 and C-3 and the presence or absence of hydroxyl groups [21,22]. The presence of two hydroxyl groups on the B ring is essential for the inhibitory effect of flavonoids on C*b*_5_R [22]. The inhibitory effect of these compounds decreases with increasing polarity of the groups attached to C-3, decreasing substantially when a hydroxyl group is attached in this position [21,22]. The development of new synthetic flavonoids with biological activity is important as they might help to understand the reactivity of flavonoids as possible ligands for enzymes [20].

Anthocyanins are a subgroup of flavonoids responsible for most red and blue colors found in flowers and fruits [23]. Although usually represented by the flavylium cation form, these species are only stable at very acidic pH values leading to the reversible formation of other species at higher pH values. Figure 2 shows the typical network of reversible chemical reactions observed for anthocyanins and synthetic flavylium salts. These dyes are usually isolated as flavylium salts. When dissolved in slightly acidic or neutral media, the red-shifted quinoidal base (A) is rapidly formed by proton transfer. However, for most compounds, these species are not the most stable, disappearing through the slower hydration of the flavylium cation to give the colorless hemiketal (B) and the *cis*-chalcone (Cc). Finally, the *cis*-chalcone may undergo a slow isomerization reaction to give the respective *trans*-chalcone (Ct). The composition of the final equilibrium strongly depends on the nature and position of the substituents decorating the flavylium core and on medium conditions such as pH, temperature, polarity, etc. This study aims to better understand the molecular mechanism of the interaction between flavonoids and C*b*_5_R by using flavylium salts as dynamic multistate compounds to design better inhibitors of the enzyme in the future.

## 2. Results

### 2.1. Structural Interconversion of Flavylium Compounds

Five flavylium salts belonging to the anthocyanin analogs subgroup were selected for screening against the ferricyanide reductase activity of C*b*_5_R to study and select the most potent inhibitors based on the percentages of inhibition (Figure 1). Flavylium cations 1, 2, 4, and 5 were available from previous studies [24,25,26]. Compound **3** was synthesized in a single step from commercially available building blocks using trimethylsilyl chloride as an acid source to generate gaseous HCl in situ.

The stock solutions of the flavylium compounds were prepared in dimethyl sulfoxide (DMSO): H_2_O (99:1) containing 0.1M of HCl. To estimate the predominant species of the used compounds (Figure 1), 10 µL of stock solution of each compound was added to potassium phosphate buffer at pH 7.0. The UV-vis spectra were recorded immediately (Appendix A, black lines) and after 2 min (red lines). No substantial spectral variations in this time scale were observed, demonstrating negligible interconversion between the species. The spectral signatures, i.e., the observation of absorption bands in the visible and UV-vis regions, pointed out the existence of the quinoidal base and colorless hemiketal/chalcone species mixtures under non-equilibrated conditions.

### 2.2. Flavylium Salts Inhibit Cb_5_R Activities

A pre-screening with a fixed concentration of each flavylium salt (20 µM) was performed to identify the most potent inhibitors for the ferricyanide reductase activity of C*b*_5_R (Figure 2a). The kinetics of the ferricyanide (500 µM) reduction by C*b*_5_R in the presence of NADH 150 µM at pH 7.0 can be compared in the absence (black squares) and the presence of compounds 1, 2, 3, 4, and 5 (labeled in red, orange, yellow, light green, and purple, respectively; Figure 2a). The activity inhibition of each compound on this activity with respect to the control (None: 340 ± 15 μmol/min/mg) was as follows: compound **1**: 36 ± 2; compound **2**: 60 ± 14; compound **3**: 143 ± 8; compound **4**: 27 ± 8; compound **5**: 5 ± 3 μmol/min/mg.

To further characterize the inhibitory potency of those compounds to strongly inhibit NADH: ferricyanide reductase activity of C*b*_5_R (compounds 1, 4, and 5), we calculated their IC_50_ values The titration of the NADH: ferricyanide reductase with increasing concentrations of each compound is found in Figure 2b. Compounds 1, 4, and 5 presented the following IC_50_ values of 2.14 ± 0.15 µM, 0.82± 0.019 µM, and 0.18 ± 0.03 µM, respectively (Figure 2b and Table 1).

Compounds **1** and **5** were selected to investigate the dependence of the IC_50_ value with their respective quinoidal base or *trans*-chalcone species. Stock solutions of compounds at acid pH values (non-equilibrated (NE) in the media) and by incubation in phosphate buffer at pH 7.0 for 24 h were prepared, and their UV-vis was measured in phosphate buffer pH 7.0. As expected, the mole fraction of the quinoidal base of compounds **1** and **5** partially disappeared, leading to the formation of the respective *trans*-chalcone species after a 24 h incubation in phosphate buffer at pH 7.0 (Appendix A). We monitored the spectra of compounds **1** and **5** (black and grey line, respectively) after immediate addition into phosphate buffer pH 7.0 (continuous line) or after incubation in this buffer for 24 h (dotted line). Our data indicate that ca. 40% of the quinoidal base (relative to its value after a 2 min incubation, λ_max_ > 500 nm) disappeared, forming the *trans*-chalcone (λ_max_ ~ 380 nm) after incubating the molecules in phosphate buffer for 24 h.

The variation of the IC_50_ value for the NE and E forms compounds **1** and **5** against the NADH: ferricyanide reductase activity of C*b*_5_R is shown in Table 2. The calculated IC_50_ values for the NE solutions of compounds **1** and **5** were: 2.14 ± 0.15 and 0.18 ± 0.03 µM, and for the E forms: 4.25 ± 0.83 µM and 2.12 ± 0.50 µM, respectively (Table 2). Our results indicate that the inhibitory properties of these compounds, including the IC_50_ values, might shift depending on the predominant species present in the solution, suggesting that the quinoidal base species are the species inhibiting the enzyme or they are more potent inhibitors of the reductase than the *trans*-chalcone species.

C*b*_5_ is the endogenous natural substrate of C*b*_5_R in biological samples. To further assess and compare the inhibitory properties of compounds **1** and **5** in non-equilibrated (NE) and equilibrated (E) conditions, we measured their effect on the NADH: C*b*_5_ reductase activity of C*b*_5_R (Figure 2c) at 20 µM concentration, in the presence of human soluble C*b*_5_ (5 µM), purified human soluble C*b*_5_R reductase (0.2 μg/mL) and NADH (150 μM). The NE forms of compounds **1** and **5** achieved an activity inhibition of 90.9 ± 1.5% and 99.3 ± 0.1%, respectively (Figure 2c), while the E forms of compounds **1** and **5** achieved 73.2 ± 1.2% and 98.9 ± 0.1%, respectively. 

### 2.3. Type of Inhibition of the NADH: Ferricyanide Reductase Activity

We determined the inhibitory mechanism of compounds **1** and **5** against the NADH: ferricyanide reductase activity of C*b*_5_R (0.35 μg/mL) by titrating this activity against the NADH concentration in the presence of compound **5** (NE). As shown in Figure 3a, the measured *V*_max_ for this activity was 95 ± 9 µM/min (open squares), which decreased in a concentration-dependent manner to 74 ± 2 and 53 ± 1 µM/min, in the presence of 0.015 μM (open circles) and 0.03 μM (open triangles) of compound **5**, respectively. To better shed light on the type of inhibition, we performed a Lineweaver-Burk data analysis (Figure 3b). A similar *K*_m_ value was obtained from the performed measurements in the absence and presence of the NE species of compound **5**: 26 ± 1 µM of NADH and different *V*max values as reported by line fitting of the data (red line) cutting at the *x*-axis and the *y*-axis, respectively. This result suggests that the NE species of compound **5** act as a non-competitive inhibitor of the NADH: ferricyanide reductase of C*b*_5_R. In addition, based on our data, the calculated inhibition constant value (*K*_i_) for the NE species of compound **5** was 0.04 µM.

### 2.4. Determination of the Dissociation Constant of the Non-Equilibrated and Equilibrated Forms of compounds **1** and **5** to Cb_5_R

Analysis of the dissociation constant of the NE and E forms of compounds **1** and **5** for the complex formation with the reductase was determined by tracking the changes in the intensity of the C*b*_5_R FAD’s autofluorescence (Figure 4), as previously performed for other ligands of the reductase [7]. 

Data fitting allowed us to measure a dissociation constant of 31.2 ± 4.7 nM and 28.0 ± 2.2 nM, for the respective NE and E forms of compound **1** and 29.2 ± 5.0 nM and 19.1 ± 1.5 nM, for the respective NE and E forms of compound **5**.

### 2.5. Molecular Docking for the Creation of In Silico Models for the Complex Formation between the Quinoidal Base (A) and trans-Chalcone (Ct) of compounds **1** and **5** with Cb_5_R

Using Autodock Vina [27], we created and selected a model for the complex formation between C*b*_5_R (PDB:1I7P) and the quinoidal base (A) and *trans*-chalcone (Ct) species of compounds **1** and **5**. The interacting site for each compound was defined as the amino acid residues that contacted the previously indicated species with a spacing of 0.35 Å and radii of 0.06 Å using AutodockTools 1.5.6 [28].

Amino acid residues of the reductase that contacted the A and Ct species of compounds **1** and **5** are shown in Table 3. Gly179 and Pro275 side chains contact with A and Ct of both compounds (Table 3 and Figure 5a,b, blue labeled). Met278 and Phe251 side chains were found to contact the A form of compound **1** (Table 3 and Figure 5a,b, red labeled). In addition, Met278 and Phe251 side chains were found to contact the A and Ct species of 5 (Table 3 and Figure 5a,b, red labeled). One amino acid side chain (Ala282) was found to contact the A species of compound **1** (Table 3 and Figure 5a,b, yellow labeled). Two amino acid side chains (Tyr 112 and Gln210) were found to contact the Ct species of compound **5** (Table 3 and Figure 5a,b, green-labeled). We also found the formation of an H-bond between the imine group (HN1 atom) of the glutamine 210 residue, involved in the peptide bond, with the hydrogen atom of the hydroxide group located in carbon four of the trans-chalcone species of compound **1**. Sequence alignment of rat PDB structure 1I7P linear sequence with the sequence of human C*b*_5_R showed no differences between isoforms concerning residues forming these isoforms (Appendix A).

### 2.6. Compounds **1** and **5** Inhibit NADH-Dependent Activities of the Cb_5_R in Microsomes

To further assess the biological relevance of the tested compounds as inhibitors of the C*b*_5_R, we measure the inhibition dependence of the NADH: ferricyanide reductase activity in rat liver microsomes (Figure 6a). All tested compounds follow a dose-dependent effect on this activity, although the maximum inhibition achieved for them was not the same as measured with the purified recombinant enzyme. The E forms of compound **5** achieved a maximum inhibition of 81 ± 2%, while the NE forms achieved a 77 ± 2% up to a compound concentration of 100 µM. An 85 ±2% inhibition was found for the E forms of compound **1**, while the NE forms of compound **1** non-significative inhibited 11 ±7% of the NADH: ferricyanide reductase activity. We also determined the IC_50_ values for each compound against the NADH: ferricyanide reductase activity of microsomes (5.6 µg/mL of protein) to obtain the following values: ND, 18.0 ± 3.8 µM, 8.0 ± 2.2 µM, and 3.3 ± 0.4 µM. for the NE and E forms of compound **1** and the NE and E forms of compound **5**, respectively.

Since in biological membranes, enzymatic activities alternative to C*b*_5_R exist using ferricyanide as a substrate, we further assessed the inhibitory effect of compounds **1** and **5** in the reductase activity of the C*b*_5_R using C*b*_5_, as a natural and more specific substrate of the C*b*_5_R, in the presence of NADH. We measured the inhibition of the NADH: C*b*_5_ reductase activity with a fixed concentration of the compounds (Figure 6b) and compared the values with those of the NADH: ferricyanide reductase activity in microsomes (Figure 6a). In the presence of C*b*_5_, the NE and E forms of compound **1** (40 µM) inhibited this activity by 25.0 ± 5.3% and 58.8 ± 1.5%, respectively; and for the NE and E forms of compound **5** by 65.6± 2.9 and 80.6 ± 0.8%, respectively. The inhibition of the NADH: ferricyanide reductase activity of microsomes using the same concentration of each of the tested compounds: 4.6 ± 4%, 63.4 ± 3.6% for the NE and E form of compound **1**, and 71.0 ± 4.7% and 81.8 ± 2.3% for the NE and E form of the compound **5**, correlated with those of the NADH: C*b*_5_ reductase activity.

Finally, we measured the effect of compounds **1** and **5** on the previously described NADH-dependent superoxide anion production of the C*b*_5_R by quantifying the inhibitory effect of the compounds on the NADH: nitro blue tetrazolium (NBT) reductase activity of C*b*_5_R from microsomes sensitive to superoxide dismutase (SOD) and catalase (CAT) (Figure 6c). The NADH: NBT reductase activity of C*b*_5_R measured with microsomes (5.2 µg/mL) was 4.4 ± 0.1 µM/min. Quantification of the inhibition of the NADH: NBT reductase activity of microsomes sensitive to SOD and CAT allowed us to measure a superoxide anion production by the rat microsomes of 2.3 ± 0.1 µM/min (52.2% of the total activity). The superoxide anion production dropped to 1.7 ± 0.1 and 1.4 ± 0.1 µM/min for the non-equilibrated and equilibrated forms of compound **1** and to 1.4 ± 0.1 and 2.4 ± 0.1 µM/min for the NE and E forms of the compound **5**, respectively.

## 3. Discussion

C*b*_5_R is an enzyme that participates in liver drug detoxification, mainly within microsomal membranes [5]. We have also described that C*b*_5_R is the main component of the plasma membrane NADH: oxidase activity of neurons that has been implicated in the NADH-dependent superoxide anion production in neuronal apoptosis [10,15,29]. The activity of the enzyme is modulable by C*b*_5_ levels as a natural ligand of the C*b*_5_R, and the superoxide anion production can be stimulated by C*b*_5_R complexation with cytochrome *c* and other substrates*,* although this enzyme can also induce superoxide anion radical production in the absence of other electron acceptors rather than oxygen [1,6,15].

Flavonoids are a family of natural polyphenols with antioxidant properties and potential use as a pharmacological drug for treating diseases, including neurodegenerative diseases [18,19]. Some flavonoids can inhibit the plasma membrane NADH oxidase of neurons blocking the associated superoxide anion production of cerebellar granular cells undergoing apoptosis in culture and the observed oxidative stress in rat models of neuropathies [30,31,32]. In addition, some flavonoids are inhibitors of the NADH: C*b*_5_ reductase and NADH: ferricyanide reductase of C*b*_5_R [22], which suggests that part of the superoxide anion production described to be produced in animal models of neuropathies and blocked by flavonoids could be ascribed to the production by this reductase [30,31,32]. On the other hand, flavylium salts which comprise anthocyanins dyes belonging to the flavonoid family, display a complex network of interconverting chemical species that can be modulated by the pH of the media [20,23]. In this work, we selected a group of synthetic flavylium salts based on the previous description of the inhibitory profile of flavonoids as inhibitors of C*b*_5_R to shed light on the fact that the flavonoid inhibitory behavior can be dependent on the predominant flavylium species present in solution, which mole fraction distribution can be modulable by medium conditions and external stimuli.

A prescreening of the inhibitory effect on the NADH: ferricyanide reductase activity of C*b*_5_R that was later assessed with the NADH: C*b*_5_ reductase activity of the protein allowed us to select two main inhibitors based on the low IC_50_, the compound **1** and **5**. We determined that both the NE and E-species of these compounds could inhibit the reductase, although the NE forms were more effective in terms of lower IC_50_ for the inhibition. In these terms, the effect of compound **5** should be emphasized. Interestingly, besides the difference found in terms of IC_50_ between compounds **1** and **5**, which is more than one order of magnitude between the NE forms (2.14 ± 0.15 µM and 0.18 ± 0.03 µM for compounds **1** and **5**, respectively), the calculated dissociation constant for all the tested compounds was, in the low nanomolar range (≈15–25 nM).

Two compounds, 1 and 5, present a certain degree of homology by the presence of one hydroxyl group at carbon position seven of the A-ring and one or two hydroxyl groups at carbon three and four of the B-ring, respectively. The presence of these hydroxyl groups at these positions in compounds **1** and **5** contrast with their relatively high inhibition of NADH: ferricyanide activity of C*b*_5_R with respect to other flavylium salts. For this reason and by comparison, it might be proposed that the presence of two hydroxyl groups at carbon position seven of the A-ring and four of the B-ring are key for inhibiting C*b*_5_R. In addition, one additional hydroxyl group at carbon three of the B-ring seems to increase the inhibitory properties. Compound **4** also presented a relatively high inhibition of the NADH: ferricyanide activity of C*b*_5_R but no described flavonoids with similar minimal structural homology has been tested up to our knowledge.

Molecules with the same characteristics in the group of flavonoids used as inhibitors of C*b*_5_R were not found [22]. Noteworthy, flavonoids presenting a hydroxyl group at carbon four of the B-ring and at least the hydroxyl group at carbon seven of the A-ring are inhibitors of C*b*_5_R activity: Naringenin did not inhibit C*b*_5_R as shown by in vitro studies. An IC_50_ of 36 µM for apigenin against NADH: ferricytochrome *b*_5_ reductase was found. Flavonoids presenting an additional hydroxyl group at de carbon four of the B-ring show higher inhibitory properties such as luteolin, quercetin, catechin, epicatechin, and taxifolin with IC_50′_s of 0.11 µM, 1.1 µM, 4.5 µM, 3.2 µM, N.D. [22]. This extra hydroxyl group is present in our compound **5** that have a lower IC_50_ value (IC_50_ 0.18± 0.03 µM) concerning compound **1** (IC_50_ 2.14 ± 0.15 µM), does not have a hydroxyl group at de carbon three of the B-ring. Our data suggest that compound **5** in the non-equilibrated form is at least as efficient as the flavonoid luteolin as an inhibitor of C*b*_5_R.

By a competitive assay against NADH, as the natural electron donor of the reductase, we determined that the inhibition mechanism of the NE form of compound **5** was non-competitive. Docking analysis against the structural model of the reductase confirmed this point since, in comparison to NADH, the tested compounds exhibited different interacting sites [1]. The analysis of the interaction between C*b*_5_R’s residues with compounds highlights the presence of Gly179 and Pro275 in both the A and Ct species of compound **1**. Met278 and Phe251 residues were present in both the A and Ct species of compound **5**. When the analysis is made in relationship with residues that contact the A or the Ct species, we found that Met278, Phe251, and Ala282 were present in the A species of compounds **1** and **5**, while Gln210 was present in the Ct species of compound **1** and **5**. Finally, we assessed the biological relevance of our findings by testing the action of compounds **1** and **5** against the C*b*_5_R present in rat liver microsomes. The weak inhibition found for the NE form of compound **1** against the NADH: ferricyanide reductase, later confirmed by the measurement of the NADH: C*b*_5_ reductase of microsomes, contrasted with the inhibitory profile of E species of compound **1** and the same behavior of the NE and E species of compound **5**. We attributed this result to the higher hydropathy index of the NE species rather than the E ones.

On the other side, it was confirmed that the tested compounds inhibited the NADH-dependent superoxide anion production of microsomes. The most potent inhibitory compound was the E form of compound **5** which blocked the superoxide anion production of the membranes at similar levels to those raised by the presence of SOD/CAT in the assay (2.3 ± 0.1 µM/min by 5.2 µg/mL of microsomes).

We conclude that flavylium salts can constitute interesting flexible tools for developing inhibitors against the C*b*_5_R that can be modulable by medium conditions and external stimuli due to their multistate properties. This family of compounds opens the door to developing modified molecules with higher specificity and susceptible to being activated under pH-dependent variables such as acid organelles or conditions such as cellular acidosis as that observed in many pathological conditions.

## 4. Materials and Methods

### 4.1. Reagents

All reagents used in this work were of maximum grade from Sigma-Aldrich or Merck.

### 4.2. Synthesis

Flavylium salt **3** was synthesized as described by Al Bittar et al. for other flavylium salts [33]: salicylaldehyde (62 mg, 0.51 mmol) and 4′-4ydroxy-3′,5′-dimethoxy acetophenone (100 mg, 0.51 mmol) were dissolved in a 3 mL of ethyl acetate: methanol (2:1, v:v). After carefully adding 20 equivalents of trimethylsilyl chloride at room temperature, a dark red solid begins to precipitate. After stirring overnight, the solid was separated by centrifugation and thoroughly washed with ethyl acetate and diethyl ether to give the chloride salt of flavylium as a dark red powder in quantitative yield (160 mg). ^1^H NMR (400 MHz, Methanol-d4 acidified with DCl) δ 9.18 (d, J = 9.2 Hz, 1H), 8.80 (d, J = 9.2 Hz, 1H), 8.34 (d, J = 8.6 Hz, 1H), 8.26–8.19 (m, 2H), 7.95–7.85 (m, 3H), 4.07 (s, 6H). ^13^C NMR (101 MHz, MeOD) δ 175.85, 156.86, 154.12, 150.72, 150.37, 139.17, 131.43, 130.67, 124.97, 120.08, 120.04, 118.86, 110.29, 57.56. HRMS-ESI (positive mode) m/z calculated for (C_17_H_15_O_4_^+^) [M–Cl^−^]: 283.0965; found, 283.0963.

### 4.3. Purification of Cb_5_R and Cb_5_

Purification of the soluble isoform of C*b*_5_R and the human erythrocyte isoform of C*b*_5_ was performed as indicated previously shown [7,15].

### 4.4. Microsome Preparation from Rat Liver

Livers were obtained from add livitum rats. Microsomes were prepared as previously performed, with slight differences [1]. After livers extraction, they were washed and rinsed three times in 10 mM Tris-CL, 0.25 M sucrose, 1 mM EDTA, and 1 mM phenyl sulfonyl fluoride (PMSF) at a pH value of 8.1; livers were chopped and homogenized using a PYREX^®^ Potter-Elvehjem Tissue Grinder with PTFE Pestle. Buffer (3 mL) was used per gram of liver tissue. After homogenization, the lysate was centrifuged at 960× *g* for 10 min at 4 °C. The supernatant was collected and centrifuged at 9800× *g* for 20 min at 4 °C. Finally, the supernatant was centrifuged at 53170× *g* for 60 min at 4 °C. The microsomal sediment was resuspended in the same buffer and frizzed at −80 °C until use.

### 4.5. Protein Determination

Protein was determined by Bradford [34].

### 4.6. Preparation of the Non-Equilibrated and EQUILIBRATED Forms of the Flavylium Cations

The transition between the acid or non-equilibrated and the equilibrated form of the flavylium salts is slow. It occurs after incubation at the determined pH in our used conditions and buffers for days. This fact allowed us to measure the effectiveness of the compounds at pH 7.0 without structural shift at times used to measure the enzymatic activities at pH 7.0.

The salts were dissolved In acidified DMSO with 0.1M of HCl to obtain the non-equilibrated or acid form of the compounds. The equilibrated or chalcone forms of the compounds were obtained by incubation in phosphate buffer pH 7.0 for 48 h.

### 4.7. Measurement of the Activities of the Cb_5_R

The NADH: ferricyanide reductase activity was measured at room temperature in 20 mM de phosphate buffer (pH 7.0), 0.15 mM NADH, 0.5 mM ferricyanide, and C*b*_5_R (0.2 µg/mL) or microsomes (5.6 or 11.2 µg/mL), in the absence or presence of the flavylium salts, using a 10 mm quartz cuvette and a spectrophotometer (Perkin Elmer Lambda40, Perkin-Elmer, Foster City, CA, U.S.A.) with a fixed wavelength at 420 nm, for the times indicated in the figures. An extinction coefficient of ferricyanide of 1 mM^−1^ cm^−1^ [1,2,6] was used to calculate the enzymatic rates.

For the NADH: C*b*_5_ reductase activity of C*b*_5_R, the following buffer was used: 20 mM de phosphate buffer (pH 7.0), 0.15 mM NADH, 5 μM of C*b*_5_ and 0.2 µg/mL ug/mL of C*b*_5_R or 5 µg/mL microsomes, in the absence or presence of the flavylium salts, using a 10 mm quartz cuvette and a spectrophotometer. A differential extinction coefficient for the C*b*_5_ reduction at 557 nm of 16.5 mM^−1^ cm^−1^ was used to calculate the enzymatic rates.

For the superoxide anion production, the NADH: NBT reductase activity of C*b*_5_R sensitive to SOD (5.2 mU/µL) and CAT (4 µg/mL) was measured in the following buffer: 20 mM de phosphate buffer (pH 7.0), 0.15 mM NADH, 0.1 mM of NBT and 0.2 µg/mL of C*b*_5_R or 5 µg/mL microsomes, in the absence or presence of the flavylium salts, using a 10 mm quartz cuvette and a spectrophotometer. A differential extinction coefficient for the C*b*_5_ reduction at 560 nm of 27.8 mM^−1^ cm^−1^ was used to calculate the enzymatic rates when needed [15].

### 4.8. Determination of the Dissociation Constants for the Complex Formation between the Human Recombinant Cb_5_R and the Flavylium Salts

Determination of the dissociation constant for the complex formation between the different forms of the compound **1** and **5** was achieved by analyzing the change of the intrinsic autofluorescence of the flavin group of the reductase after complex formation, as previously used for other ligands of the reductase [6,7]. The fluorescence intensity of the C*b*_5_R after the sequential addition of increasing amounts of compound to phosphate buffer saline buffer (PBS) in the presence of C*b*_5_R (0.25 µM) while stirring at pH 7.0 at 25 °C was monitored using a spectrofluorometer (Photon Technology International Inc. Quantamaster, Ford, U.K.) and fixed excitation and emission wavelength at 470 and 520 nm and using a 2 mL quartz cuvette. The obtained data after titration were adjusted to the following equation for the calculation of the K_d_ for the complex formation:F − F_0_ = ([Compound] × F_max_)/(K_d_ + [Compound])
where F is the C*b*_5_R fluorescence change induced by the addition of the compounds to the cuvette, F_0_ is the initial C*b*_5_R fluorescence and F_max_ is the fluorescence increment after adding the compounds, and K_d_ is the dissociation constant value.

### 4.9. Molecular Docking Analysis

AutodockTools was used to prepare the pdbqt files based on the obtained file for the C*b*_5_R from the Protein data bank with the code PDB code 1I7P de Protein Data Bank (PDB) that corresponds to that obtained by X-Ray diffraction of the crystal structure of the rat C*b*_5_R in complex with FAD [35]. The files for the structure of the flavylium salts were prepared with the software Chemdraw 20.0 (PerkinElmer). The molecular docking analysis to study the interaction between C*b*_5_R (receptor) and the flavylium salts (ligands) was performed with Autodock Vina [27]. From all the possible conformation and clusters, the highest affinity was selected with the higher number of poses as the model for the complex formation. The interacting site for each compound was defined as the amino acid residues that contacted the previously indicated species with a spacing of 0.35 Å and radii of 0.06 Å using AutodockTools [28]. Graphics and molecular analysis were performed with the UCSF Chimera software [36].

### 4.10. Statistic Analysis of the Data

Data shown in this manuscript are the average of experiments performed in triplicate with the standard deviation of the data. Statistical analysis of the results was performed by applying the T-student analysis to determine the statistical significance of the data * *p* < 0.05, ** *p* < 0.01, *** *p* < 0.001.

## Data Availability

Data are contained within the article or Appendix A. The raw data presented in this study are available on request from the corresponding author (A.K.S.A.).

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
