# Peer review of "The Use of Flavylium Salts as Dynamic Inhibitor Moieties for Human Cb5R"

_molecules, 2022, doi:10.3390/molecules28010123_

Round 1
Reviewer 1 Report
The manuscript entitled "The use of flavylium salts as dynamic inhibitory moieties for 2 human Cb5R" describes the Cb5R inhibitory properties of flavylium salts.
The work is well-designed and presented. However, the authors should address the following before being accepted in Molecules.
1. There is no standard/reference drug to compare the data?
2. What about the logP value and any evidence of crossing BBB?
3. How the present work outcome is better than the reported ones?
Author Response
We have annexed a document addressing the reviewer's questions

Reviewer 2 Report
The paper entitled “The use of flavylium salts as dynamic inhibitory moieties for human Cb5R” by Basilio and Samhan-Arias et al describes a mechanistic study of Cytochrome b5 reductase (Cb5R) inhibition by flavonoids, in which the authors used different flavylium salts of anthocyanin family.
Five known flavylium salts were proposed by the authors as modulable chemical species to perform this work. These species, which are only stable at low pH values, undergo reversible reactions at pH values ≥ 7 providing mixtures of quinoidal, hemiketal and chalcone moieties. The structural rearrangement of the titled compounds was checked by UV-vis spectroscopy.
Firstly, the five compounds were explored in non-equilibrated (NE) conditions as inhibitors of Cb5R. The two most potent compounds were selected for a more comprehensive study, using NE and equilibrated (E) conditions, different enzyme substrates and two enzyme versions (isolated and in rat liver microsomes). The inhibition kind, the dissociation constant and the ability of the two selected compounds to reduce the superoxide anion production were also studied.
In addition, a small docking study was performed to predict the possible drug/target interactions
This paper contains a wide and interesting molecular biology study aimed at exploring the Cb5R enzyme as a drug target and fits within the scope of the journal. However, the manuscript is not carefully organized and written; is difficult to follow it and contains some mistakes. Therefore, several changes or clarifications must be carried out in the document before being accepted for publication in the journal Molecules.
Suggestions and comments:
1. Add an abbreviations list in the manuscript, including CAT, NBT, SOD etc.
2. It would be convenient to replace “NADH:...reductase activity of Cb5R” with “NADH-…reductase activity of Cb5R” throughout the document.
3. The captions of Figures 2 and 3 are excessive and must be shortened. In addition, the caption of Figure 3 is confused with the main text. In this Figure 3, there is a problem with the text extension and font size. Please clarify in both Figures the caption of panels (a, b…).
4. Check and correct the first paragraph of page 5 (lines 146-157), the text does not match the data in table 1 and Figure 2c.
5. In the results part, please explain, briefly, how the structural model for docking was constructed, also indicating the interaction types. If you use three-letter codes for amino acids, only the first letter should be uppercase.
- On page 8 (lines 269-273), check and correct the paragraph.
7. Replace “compound 9” by “compound 5” (page 8, line 279)
8. The graphic representation type of panels a) and b) in figure 4 should be the same to be able to compare results.
9. Figure 4a should be cited in the document
10. Check and correct the concentration data of studied compounds because there are differences between the results section, discussion part and figures.
11. The dissociation constant data mentioned in the discussion section do not match those in the results part.
12. On page 10, line 351, replace “hydropathy” with “hydropathy index”.
13. In line 440, check and correct the phrase.
Author Response

(The authors gave the same response as above.)

Round 2
Reviewer 1 Report
Thank you for revising the manuscript and responding to the concerns.
Author Response
The response to the reviewer's comments has been annexed.

Reviewer 2 Report
The paper entitled “The use of flavylium salts as dynamic inhibitor moieties for 2 human Cb5R” has been checked and corrected by the authors. They have made appreciable changes improving the clarity and quality of the manuscript.
However, in my opinion, there are still some minor modifications and corrections that should be performed.
- On page 3, line 92, please replace “compounds 3 was…” with “compound 3 was…”
- Figure 2, panel b, maintain the identification colour for compound 4 (light green), to avoid confusion with the control (black in panel a). Check and correct the statistical parameters p (compare captions of panel c and Figure 2c).
- Table 1, specify that the IC50 values correspond to NE species of compounds 1, 4 and 5.
- Figure S2 does not correspond to the UV-vis spectrum of compounds 1 and 5. Please, revise this.
- Please, improve the quality of Figure 5, panels a/b. Both pictures are too poor.
- Revise the amino acid residues set that establish important interactions with the A and Ct species of compounds 1 and 5. There are important differences between those mentioned in the results part and the discussion part. Also, check the codes in Figure 5 and Table 3
- Page 7, line 238, “Gln210: HN1” what does it mean?
Author Response

(The authors gave the same response as above.)
